Physical activity habits and preferences in the month prior to a first-ever stroke

McDonnell Michelle N. 1 michelle.mcdonnell@unisa.edu.au
Esterman Adrian J. 2
Williams Rosena S. 1
Walker Jenny 3
Mackintosh Shylie F. 1
1 International Centre for Allied Health Evidence, School of Health Sciences, University of South Australia , Adelaide , Australia
2 School of Nursing and Midwifery, University of South Australia , Adelaide , Australia
3 Physiotherapy Department, Flinders Medical Centre , Adelaide , Australia
Abdullah Jafri
Electronic publication date: 2014 Jul 10
Publication date: 2014
Volume: 2
Electronic Location ID: e489
Received 2014 Apr 21; Accepted 2014 Jun 26
Copyright: © 2014 McDonnell et al.
Copyright year: 2014
Copyright holder: McDonnell et al.
License: This is an open access article distributed under the terms of the Creative Commons Attribution License, which permits unrestricted use, distribution, reproduction and adaptation in any medium and for any purpose provided that it is properly attributed. For attribution, the original author(s), title, publication source (PeerJ) and either DOI or URL of the article must be cited.
License URL: https://creativecommons.org/licenses/by/4.0/

Keywords: Stroke, Primary prevention, Physical activity, Risk factor

Funding: School of Nursing, University of South Australia National Health and Medical Research Council of Australia National Stroke Foundation Research This work was funded by an internal grant from the School of Nursing, University of South Australia. Dr McDonnell was supported by a Fellowship (ID 590133) from the National Health and Medical Research Council of Australia and a National Stroke Foundation Research Grant. The funders had no role in study design, data collection and analysis, decision to publish, or preparation of the manuscript.

==============================
Background. Physical inactivity is a powerful risk factor for stroke and other chronic diseases. The aim of this study was to explore physical activity habits and preferences in the month leading up to a first-ever stroke, and to determine whether participants were aware of the link between stroke and physical activity.

Methods. We undertook an observational study with 81 participants recently admitted to a stroke unit. Participants reported their pre-morbid physical activity preferences and habits and completed the Barriers to Physical Activity and Disability Survey. Data were analysed with summative content analysis and descriptive statistics.

Results. Only 31% of participants were aware that physical inactivity was associated with stroke. Most participants defined physical activity with examples of instrumental activities of daily living (IADL) and walking (48% of responses), and IADLs constituted their most frequent regular physical activity (38% of responses). The barriers to physical activity reported by participants most frequently were lack of motivation (52%), lack of interest (50%) and lack of energy (42%).

Conclusions. Regular physical activity is important to prevent stroke and other chronic diseases but adults at risk of stroke have little awareness of the risks of physical inactivity and little motivation to undertake regular exercise.

Background

Stroke remains the leading cause of serious long-term disability, with direct and indirect costs in the United States in 2009 totalling $38.6 billion and increasing rapidly (Go et al., 2013). It is well established that a healthy lifestyle reduces the risk of stroke by up to 80%, and undertaking regular physical activity (PA) is an important component of this (Chiuve et al., 2008). Physical inactivity is a powerful risk factor for stroke, with a large multinational case-control study demonstrating that inactivity is the second greatest risk factor for stroke following hypertension (O’Donnell et al., 2010). Despite this, interventions to improve uptake of adherence to PA recommendations have received little research attention, with emphasis instead on controlling risk factors such as hypertension, body mass index, cholesterol and diabetes. While this has demonstrated significant impact on stroke reduction (Kahn et al., 2008), there is still a pressing need to do more to reduce stroke incidence and mortality.

Although the precise amounts and type of exercise required to prevent stroke are unclear, meta-analyses conclude that regular PA reduces the risk of stroke by 25–30% when compared with the least active people (Goldstein et al., 2011; Lee, Folsom & Blair, 2003). In a telephone survey of residents in the US state of Ohio, only 11% of the 2,173 respondents were aware that lack of exercise was a risk factor for stroke (Schneider et al., 2003), with more recent data from Ireland showing a higher proportion (32.5%) but still generally a low awareness (Hickey et al., 2012). Physical inactivity is occasionally even overlooked by researchers considering important modifiable risk factors for stroke when studying awareness of stroke in the community (Kleindorfer et al., 2009).

Physical activity levels in older adults tend to decline with age, with 40% of US adults aged 65–74, and 53% of those aged ≥ 75, being classified as inactive (not undertaking any light/moderate or vigorous activity of at least 10 min per day) (Schiller et al., 2012). Older adults frequently report poor health, lack of company and lack of interest as the main barriers to PA (Moschny et al., 2011). Increasing our awareness of the barriers and facilitators to exercise in stroke-free individuals at risk of stroke will enable the targeting of health promotion activities to these groups, particularly those with hypertension, diabetes and previous transient ischaemic attack (TIA).

The purpose of this study was to interview people recently admitted to the hospital with a first-ever stroke to explore their PA patterns prior to their stroke, whether they were aware of the link between stroke and PA and their understanding of what PA involves. A further aim was to examine barriers and facilitators to PA, to highlight which factors may be modifiable and could be addressed to prevent stroke.

Methods

Study population

This cross-sectional study recruited adults with first-ever stroke from the acute stroke units of two local metropolitan hospitals and included quantitative and qualitative components. Participants were included if they had a first-ever stroke and were aged 40–90 years. Participants were excluded if they had a history of dementia, had receptive and expressive language difficulties which would impact upon their ability to take part in the interview, and were drowsy or otherwise unable to cooperate. All participants gave their informed written consent for the study which was approved by the relevant Research Ethics Committees (the University of South Australia, Royal Adelaide Hospital and Southern Adelaide Clinical Human Research Ethics Committee).

Protocol

Eligibility criteria were verified from medical records. Interviews were conducted on the hospital ward, within a month of their stroke. First, three open ended questions were posed to understand how people with stroke view the relationship between PA and stroke risk, as previously used to explore attitudes towards PA in people with hypertension (Beune et al., 2010). This method allows participants to share their experiences from their own point of view, and allows for themes to emerge based on participants’ perceptions of illness:

(1) Over the past month, did you do anything to maintain or increase your PA?

(2) How do you view the link between stroke and PA?

(3) What would you define as PA?

The interviewer prompted with specific examples if required.

The second part of the interview involved completing the Barriers to Physical Activity in Disability Survey (B-PADS) (Rimmer, Wang & Smith, 2008). The B-PADS comprises 34 items, the majority being yes/no questions designed to explore barriers (personal, environmental and/or facility related) that might influence exercise participation (e.g., have you ever been injured from exercising before? Are you concerned that exercise could make your condition worse?).This tool has been validated in people following stroke living in the community and has high reproducibility (Cohen’s κ = 0.76) and inter-rater reliability (κ = 0.86) (Rimmer, Wang & Smith, 2008).

The interview times ranged from 8 to 25 min. All interviews were digitally recorded and transcribed verbatim.

Data analysis

Data were summarised with appropriate descriptive statistics (mean and standard deviation (SD) for continuous variables, count and frequency for categorical variables). For the primary outcome, PA habits and awareness of PA as a risk factor for stroke, transcripts were examined and analysed using summative content analysis (Kondracki, Wellman & Amundson, 2002) to identify related concepts and determine frequency of these concepts within participants’ responses. This method of analysis has the advantage of being unobtrusive but also allowing interpretation of the underlying meaning of the content (Hsieh and Shannon, 2005). Two investigators (MMcD and RSW) independently generated initial codes from interview transcripts, before focusing and grouping the codes into concepts. Concepts were refined and named, and transcripts reviewed again to verify against coded data extracts. The frequency of these concepts within the sample of responses was calculated and reported as a percentage of the total number of terms coded for each question. To address the secondary aim, barriers to participating in PA in this population pre-stroke, frequency of responses to the B-PADS were summarised. Associations between variables were explored with generalized linear models using STATA® software (Version 10) and p < 0.05 was considered statistically significant.

Results

Demographic data for the 81 adults recruited are provided in Table 1. Two thirds of the participants were male, the mean age was 67 years and they were interviewed approximately one week after their first-ever stroke.

Table 1 Demographic characteristics of participants.

	n = 81	
Sex	54M, 27F	
Age (mean, SD)	67.6 ± 13.2	
Days since stroke (median, range)	6 (1–30)	
Stroke type, infarct (n, %)	56 (69%)	
Stroke location, cortical stroke (n, %)	44 (54%)	
Stroke risk factors (n, %)		
Hypertension	53 (65%)	
Diabetes	16 (20%)	
Current smoker	21 (26%)	
Hypercholesterolaemia	28 (35%)	
Previous TIA	7 (9%)	
Cardiac complicationsa, ≥1	34 (42%)	
Number of other comorbiditiesb (median, range)	3.8 (1–10)	
Notes.

a Cardiac complications included, but were not limited to, ischaemic heart disease, myocardial infarct, coronary artery bypass grafts, heart failure, permanent pacemaker, atrial fibrillation, cardiomyopathy, congestive cardiac failure.

b Other comorbidities included, but were not limited to, depression, emphysema, rheumatoid arthritis, joint replacements, hypothyroidism, obesity, diabetes, renal failure, anaemia, chronic obstructive pulmonary disease.

Physical activity prior to stroke

Participants, who were within a month of their first-ever stroke (n = 81), were asked about their PA levels in the month prior to their stroke. There were a variety of responses, often involving more than one category, and 81% of participants reported that they did some form of PA in the month prior to their stroke. The common responses, and associated categories, are shown in Table 2.

Table 2 Physical activity levels in the month prior to stroke.

Question: Over the past month, did you do anything
to maintain or increase your physical activity?	Number of codes identified
and % of responses compared
to total number of responses	
Yes (81% participants)	Walking	43, 30%	
	IADLs e.g., shopping, work	54, 38%	
	Regular exercise e.g., gym,
running, swimming	25, 18%	
No (19% participants)	Unable to (too hot, recently unwell)
or don’t need/want to	19, 14%	

The majority of participants reported that the only regular PA they did in the month prior to the stroke consisted of instrumental activities of daily living (IADL), as mentioned by Participant #66:

“Shopping and putting away of groceries. Usually an hour, not every day, two times a week”.

Almost half of the participants reported they did some form of walking each day, and some even counted their steps:

“Walking, things like that. I don’t get up and go for a ten km walk. I just walk a lot. 9000 steps, and most days I reached that”.

Participant #40.

Only 25 of the 81 participants reported regular exercise in the month prior to their stroke, with swimming, golf, attending the gym and running given as examples.

Of those 19 participants who did not undertake any regular PA in the month prior to their stroke, many gave examples of falls, pain or other conditions which limited their activity.

“I was a slug. Because I have been having trouble with my back. I haven’t been able to do anything very much at all”.

Participant #69.

The link between stroke and PA

The responses to the question “How do you view the link between stroke and PA” were assigned a dominant response per participant. This revealed a striking lack of awareness of physical inactivity as a strong risk factor for stroke. Two participants suggested that too much PA in the days leading up actually caused their stroke. The majority of respondents (41%) had never considered the possibility that the two factors were linked and frequently used the term “no idea” (Fig. 1) and is illustrated by this quote:

Figure 1 Proportion of responses to the question “How do you view the link between stroke and PA” coded as: PA can cause a stroke (3%), they are unrelated (25%), PA reduces stroke risk (31%) or no awareness stroke of a link (no idea) (41%).

“No, I don’t know I’m not the doctor!”

Participant #7.

Many participants felt that PA was unrelated to stroke, and gave examples of stress, diet or family history as the cause for their stroke.

“I don’t think there is a link between stroke and physical activity, I don’t know about diet but smoking seems to be a bad one for me!”

Participant #4.

“I think it could, I had it in my family. My mum passed away with a stroke”.

Participant #50.

Definition of PA

Finally, in response to the question “What would you define as physical activity?” there was a lot of overlap between categories, as shown in Table 3. Again, PA was defined using examples of IADLs such as gardening, shopping and housework (47 participants) or walking (57 participants).

Table 3 Definition of PA.

Question: “What would you define as physical activity?”	Number of codes identified
and % of responses compared
to total number of responses	
1. Walking for exercise	57, 26%	
2. IADL related	47, 22%	
3. When prompted for examples then sport mostly	51, 23%	
4. Egocentric, what they had done in past months to years	26, 12%	
5. Physiological definition	37, 17%	

One in four participants reflected upon what they had done recently, for example:

“In general, everything that I do. I do my back garden, take my walker and do my walking”.

Participant #13.

A number of participants required prompting to answer this question, and this often led to mentioning sport as the definition of PA:

“Exercise and going to gym and things like that. Playing sports”.

Participant #34.

Barriers to physical activity survey

The B-PADS survey was used to collect information on the types of barriers that participants may perceive related to exercise participation. The majority of participants had exercised regularly in the past (58 participants, 72%) although many reported that they had stopped regular exercise decades ago. Few participants had been advised to exercise by their regular doctor (n = 20, 25%) and of those who were advised to exercise, only half were told to do anything specific. Despite this, 80% of participants (n = 65) felt that an exercise program could help them, and 77% of participants (n = 62) were aware of a fitness centre that they could get to. However, 31% of participants (n = 25) reported that health problems have caused them to stop exercising, and a similar proportion (32%, n = 26) had been injured from exercising in the past.

The barriers to PA reported by participants were most frequently intrinsic personal factors, as shown in Table 4.

Table 4 Self-reported barriers to physical activity prior to a first-ever stroke, according to the Barriers to Physical Activity and Disability Survey.

Barrier	% of participants
(n = 81)	
Personal		
Lack of motivation	58	
Lack of interest	50	
Lack of energy	42	
Exercise is boring or monotonous	42	
Lack of time	40	
Pain prevents me from exercising	31	
Health concerns prevent me from exercising	29	
Environmental/facility		
Cost of the program	36	
Lack of transportation	26	
Not aware of fitness centre in the area	23	
Don’t feel trainer in facility is able to help	18	

Relationship between variables

To explore the effect of participant characteristics on the likelihood of being regularly active, participants were grouped into those who did regular exercise (regular walking or other exercise such as golf, of going to the gym, 46%) and those who did not exercise regularly in the past month (54%). There was a significant inverse relationship between age and likelihood of being active, with a 30% increase in physical inactivity with each decade of advanced age (p = 0.01, Incidence Rate Ratio 1.30). There was no association between walking aid use (no aid vs. walking stick/frame/wheelchair prior to stroke) and being active (p = 0.29) or between awareness of physical inactivity as a risk factor for stroke and likelihood of being active in the month prior to stroke (p = 0.43).

Discussion

The primary outcome of this study was to understand PA preferences and the association of PA with stroke risk. We found that in the month prior to their stroke, participants were much more likely to be involved in IADL and walking than sports and other formal modes of exercise. The majority of participants had never considered the relationship between PA and stroke, and only one third were aware that inactivity was in fact a risk factor for stroke. Few participants were advised to exercise by their local doctor, despite two-thirds of the participants having hypertension and almost half suffering from at least one cardiac complaint. These results must be interpreted with caution, however, due to the retrospective nature of our study design.

This study highlights the steps that need to be taken to increase the awareness of physical inactivity as a powerful stroke risk factor, and to raise awareness of the impact that healthy lifestyle choices can have on reducing their risk of stroke. One quarter of participants had a fatalistic attitude to their own stroke, reporting that it was inevitable due to family history or because they were unlucky:

“I think it’s a roll of the dice”.

Participant #37.

When asked what physical activity they had done in the month prior to their stroke, 81% of participants responded that they were physically active. However, when questioned further it was clear that only 30% were involved in regular moderately intense exercise. In this sample of older adults, many defined PA as walking but their only regular physical activities were IADLs such as housework and gardening.

The B-PADs survey revealed that environmental barriers were not a concern for the majority of people prior to stroke. Most participants knew of a local fitness centre, and transport was not an issue. This contrasts markedly from the results of Rimmer, Wang & Smith (2008) where community-dwelling stroke survivors reported environmental/facility barriers much more frequently. A notable difference is that our participants were reflecting back to their activity levels prior to stroke. In this context, our findings are similar to the work of Moschny et al. (2011) with healthy older adults who report poor health (58%), lack of company (43%) and lack of interest (37%) as the most frequent barriers to PA.

The overwhelming majority of participants responded that intrinsic personal factors limited their desire to take part in exercise, most commonly lack of interest, motivation and energy. Some participants expressed a strong dislike of exercise, although they considered themselves to be physically active. The challenge in these individuals is to engage them in activities that they are interested in, focusing on the social aspects and the possibility that physical activity can be enjoyable for physical health and mental well-being (Moschny et al., 2011), and move away from structured regimes involving limited exercise options. This is particularly important given the 30% decrease in PA for each decade of advanced age: finding physical activities that are safe and interesting for older adults to participate in may help prevent the age-related decline in PA.

Although many participants did suffer from cardiac conditions and other co-morbidities such as asthma, diabetes and arthritis, less than a third of participants reported that health concerns prevented them from exercising. This may in part be due to the poor awareness of the role of physical activity in managing and improve conditions such as diabetes and cardiovascular disease. The challenge for primary prevention is to encourage people to take up regular exercise, particularly if they do present with other stroke risk factors in the hope that early intervention can prevent later stroke and profound disability. Even those who have not participated in exercise recently can be reassured that in the event of a stroke, higher PA levels in the month prior to stroke are associated with better short-term functional outcomes after stroke (Stroud et al., 2009).

A limitation of this study was that we did not quantify the amount or intensity of physical activity that was performed in the month prior to the stroke. This could have been done to ascertain whether participants met the recommended Physical Activity guidelines (Department of Health and Aged Care, 1999), or quantified their PA in terms of metabolic units per week using the International Physical Activity Questionnaire (Hagstromer, Oja & Sjostrom, 2006). This is an inherent limitation of the retrospective design of the study, and reports of PA may be influenced by the stroke event itself. This was not completed partly because of the likelihood of recall bias but also to avoid any distress that may be caused by such questioning so close to their stroke event. Future studies should address this, because of the known link between moderately vigorous physical activity and reduced risk of stroke (McDonnell et al., 2013). However, a strength of this study was the relatively large sample size for a qualitative study, allowing insights into physical activity habits which may not have been captured with standardised surveys.

Conclusions

Adults who have recently suffered a stroke report little motivation to undertake exercise, and little awareness of the risk of stroke from low levels of physical activity. They define physical activity using low intensity activities like shopping and walking, and these are the most common forms of PA that they undertake. The challenge for future health care policy is to increase awareness of PA as a modifiable risk factor for stroke, and engage older adults in suitable ways to increase their PA, particularly moderately intense activity in accordance with PA guidelines.

Supplemental Information

Supplemental Information 1 Ethical approval confirmation

Click here for additional data file.

Supplemental Information 2 Interview and BPADS data

Interview and BPADS dataset.

Click here for additional data file.

Additional Information and Declarations

Competing Interests

Author Contributions

Human Ethics

Jenny Walker is an employee of Flinders Medical Centre.

Michelle N. McDonnell conceived and designed the assessments, performed the assessments, analyzed the data, wrote the paper, prepared figures and/or tables, reviewed drafts of the paper.

Adrian J. Esterman conceived and designed the assessments, analyzed the data, reviewed drafts of the paper.

Rosena S. Williams performed the assessments, reviewed drafts of the paper.

Jenny Walker performed the assessments, contributed reagents/materials/analysis tools, reviewed drafts of the paper.

Shylie F. Mackintosh conceived and designed the assessments, performed the assessments, analyzed the data, reviewed drafts of the paper.

The following information was supplied relating to ethical approvals (i.e., approving body and any reference numbers):

Human Research Ethics Committee, University of South Australia.

Ethical approval was confirmed by letter on 31/10/2011.

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
