# Peer review of "Physical activity habits and preferences in the month prior to a first-ever stroke"

_PeerJ, doi:10.7717/peerj.489_

## Round 0.1 · original submission · Minor Revisions

Dear Authors,Thank you for submitting this manuscript which will need minor revisions.One reviewer has attached an annotated manuscript which will help you do the corrections .The revised manuscript when re-submitted will undergo re-review.

Reviewer 1 ·

Basic reporting

Language needs minor correction / small sentences at many places.
Figure provided is not essential

Experimental design

Design in terms of qualitative study needs to be redefined
comments are also mentioned attached PDF

Validity of the findings

some of the discusion is made in contrast to the subjective knowledge

Additional comments

the research question could have been solved using more objective methodology

Annotated reviews are not available for download in order to protect the identity of reviewers who chose to remain anonymous.

·

Basic reporting

good. some areas need more details like the interview guide (even though a reference is given). the theory on which the qualitative data was analysed- could make it more clearer to read.

Experimental design

A retrospective cohort was chosen, however the limitation of a retrospective interview has not been discussed. It would make sense to add them.

Validity of the findings

Conclusion is solid. Again because of the retrospective nature of questioning the data should be viewed with caution - due to underestimating or overestimation have not been clearly discussed.

Additional comments

Good work which could have implication in prevention literature.

Reviewer 3 ·

Basic reporting

The article was well and clearly written with clear background and clear objective of the study.

Experimental design

The research question was well define and was answered via interview.

Validity of the findings

The findings are acceptable.

Additional comments

The study is very relevant to the current needs in preventing or reducing stroke incidence.

---

## Round 0.2 · accepted · Accept

Dear Authors,Thank you for the revised manuscript.The revised manuscript will now undergo the next process as it is now deemed suitable for publication in Peer J by the reviewers,

Reviewer 1 ·

Basic reporting

the manuscript is now acceptable

Experimental design

no comments

Validity of the findings

no comments

Additional comments

the manuscript is now acceptable